# Suffruticosol C-Mediated Autophagy and Cell Cycle Arrest via Inhibition of mTORC1 Signaling

**DOI:** 10.3390/nu14235000

**Published:** 2022-11-24

**Authors:** Senlin Qin, Huijun Geng, Guoyan Wang, Lei Chen, Chao Xia, Junhu Yao, Zhangzhen Bai, Lu Deng

**Affiliations:** 1College of Animal Science and Technology, Northwest A&F University, Xianyang 712000, China; 2College of Landscape Architecture and Arts, Northwest A&F University, Xianyang 712000, China

**Keywords:** suffruticosol C, anticancer therapy, mTORC1, autophagy, cell cycle arrest

## Abstract

*Paeonia* species are well-known ornamental plants that are used in traditional Chinese medicines. The seeds of these species are rich in stilbenes, which have wide-ranging health-promoting effects. In particular, resveratrol, which is a common stilbene, is widely known for its anticancer properties. Suffruticosol C, which is a trimer of resveratrol, is the most dominant stilbene found in peony seeds. However, it is not clear whether suffruticosol C has cancer regulating properties. Therefore, in the present study, we aimed to determine the effect of suffruticosol C against various cancer cell lines. Our findings showed that suffruticosol C induces autophagy and cell cycle arrest instead of cell apoptosis and ferroptosis. Mechanistically, suffruticosol C regulates autophagy and cell cycle via inhibiting the mechanistic target of rapamycin complex 1 (mTORC1) signaling. Thus, our findings imply that suffruticosol C regulates cancer cell viability by inducing autophagy and cell cycle arrest via the inhibition of mTORC1 signaling.

## 1. Introduction

*Paeonia* species, which are commonly known as ornamental plants, are also used in traditional Chinese medicine. In the last two decades, studies have shown that the seeds of peonies are abundant in functional fatty acids, includingα-linolenic acid, linoleic acid, and oleic acid, which has various health-promoting benefits [1,2,3]. Lately, there has been increasing interest in the use of peony seed oil as a health supplement, and the Chinese Ministry of Health even approved it as a novel food. In addition to nutritional unsaturated fatty acids, there are other bioactive constituents in peony seeds that have attracted the attention of researchers. Recently, peony seeds were found to be rich in stilbene compounds. Trans-resveratrol (trans-3,4′,5-trihydroxistilbene), the most common stilbene compound, has wide-ranging health-promoting properties, including antioxidant, anticancer, anti-atherosclerotic, anti-diabetic, estrogenic, anti-osteoporosis, anti-inflammatory, neuroprotective, cardioprotective, anti-aging, and anti-obesity effects [4,5]. Up until now, over 20 stilbenes have been reported in the species of genus, *Paeonia*, most of which are glycosylated, methylated, and oligomeric forms of resveratrol [6,7,8,9]. Suffruticosol C, a trimer of resveratrol, is one of the most dominant stilbenes in peony seeds that may have potential health benefits, and this needs to be explored further [10,11].

It has been reported that resveratrol has antitumor potential in various human tumor models [12], and its antitumor effect is linked with its ability to regulate autophagy and the cell cycle progression of cancer cells. The cellular process known as autophagy, which removes damaged organelles or cellular components, draws our interest due to its strong correlation with many diseases, including cancer [13,14]. In the context of autophagy, the mechanistic target of rapamycin complex 1 (mTORC1) received substantial research interest [15]. In detail, some early autophagy associated proteins, such as autophagy-related protein 13 (ATG13) and unc-51-like kinase 1 (ULK1), can be phosphorylated and inactivated by mTORC1 [16,17,18,19]. Furthermore, two transcription factors, EB and E3 (TFEB and TFE3), are vital elements regulating the lysosome biogenesis-related genes and the autophagosome initiation [20]. Cancer is a group of diseases of uncontrolled cell division, which is tightly regulated by a series of multiple cell cycle regulators [21]. The cell cycle is divided into four sequential phases. The most important phases are the S phase, the period of DNA replication, and the M phase, the period when the cell divides into two daughter cells. The gap between the S and M phases is called the G2 phase. The G1 phase is the onset of mitosis, the period in which cells sense positive and negative signals from the growth signaling network. Cells in the G0 phase are only temporarily out of the cell cycle. Once instructed by signals, cells in the G0 phase will quickly return to the cell cycle [21]. The progression of the cell cycle in cancer cells is primarily regulated by three protein families: cyclins, cyclin-dependent kinases (CDKs), and CDK inhibitors (CDKIs) [22]. However, changes can occur in cells including cyclin amplification, CDK mutation, and the inactivation of tumor suppressor genes, leading to aberrant CDK activities, the amplification of growth signals, and the loss of control over cellular growth [23]. Recently, evidence suggested that the health-promoting properties of resveratrol are attributed to its ability to induce autophagy and cell growth arrest [12,24]. Although there is evidence linking resveratrol directly with autophagy and the cell’s cycle, it remains unclear whether suffruticosol C has an antitumor effect or whether it has a regulatory effect on cancer cell viability. Therefore, in the present study, we intend to determine the anti-cancer effects and mechanisms of suffruticosol C in various cancer cell lines. With regard to the molecular mechanisms, we tried to investigate whether the effect of suffruticosol C is regulated via mTORC1. This is because mTORC1 is a key regulator of autophagy and cell cycle progression [20,25,26], and some studies have reported that resveratrol may negatively regulate mTORC1 activation [27,28,29,30]. However, the biological function of suffruticosol C against cancer is still unclear, let alone the underlying mechanisms.

In the current study, we propose determining the anti-cancer effects and cell death mechanisms of suffruticosol C in various cancer cell lines. We found a dose-dependent manner of cell death when there was exposure to suffruticosol C. Furthermore, the treatment of cells with suffruticosol C induced autophagy and cell cycle arrest, suggesting that autophagy and the cell’s growth arrest was involved in suffruticosol C-associated anti-cancer effects. Finally, we show that suffruticosol C triggers autophagy and cell growth arrest by inhibiting mTORC1 signaling.

## 2. Materials and Methods

### 2.1. Reagents

Suffruticosol C and resveratrol were isolated from the seeds of the tree peony. The concentration was chosen based on previous studies [27,28,31,32] and considering that suffruticosol C is a trimer of resveratrol. The secondary antibodies and Erastin (E7781) were provided by Sigma-Aldrich (St. Louis, MO, USA). RSL3 (1219810-16-8) was obtained from Cayman (Ann Arbor, MI, USA), and Bafilomycin A1 (S1413) was obtained from Selleck Chemicals (Houston, TX, USA). PBS and trypsin were obtained from HyClone (Logan, UT, USA). The RPMI 1640 and DMEM media, streptomycin, β-mercaptoethanol, penicillin, and fetal bovine serum (FBS) were purchased from Gibco (Grand Island, NY, USA). The TB Green qRT-PCR kit (RR820A), PrimeScript™ RT reagent Kit (RR047A) and TRIzol were provided by Takara (Dalian, China). The primary antibodies used are listed as follows: Bcl-2 (12789-1-AP), Bax (50599-2-Ig), LC3II (18725-1-AP), p62 (18420-1-AP), and Actin (20536-1-AP) were purchased from Proteintech (Rosemont, IL, USA); S6 (2217S), p-S6 (4858S), AKT (9272), pT389-S6K (9234S/L), S6K (9202S), and Cleaved Caspase-3 (D175) were purchased from Cell Signaling Technology (Danvers, MA, USA).

### 2.2. Cell Culture

HT29, H1299, Caco2, HepG2, HCT116, and PC3 cells were purchased from National Science and Technology Infrastructure (NSTI, Shanghai, China). H1299 and HT29 were cultured in the RPMI 1640 medium with 10% FBS, while Caco2, HCT116, and HepG2 were cultured in the DMEM medium with 10% FBS following the ATCC guidelines.

### 2.3. Quantitative RT-PCR Analysis

TRIzol reagents were used to isolate total RNA. RNase-free Dnase was added to remove DNA. The RNA was then reverse transcribed. Sequentially, qRT-PCR analysis was conducted in technical triplicate in a Roche LightCycler^®^96 qRT-PCR system (Roche, Penzberg, Germany) to determine the relative levels of target mRNA [33]. The primers of the target genes used are listed in Table 1.

### 2.4. EdU Assay

The EdU (5-ethynyl-2-deoxyuridine) assay protocol was conducted as outlined in a previous study [34]. The Click-iT EdU Cell Proliferation Kit (C10337) was obtained from Thermo Fisher Scientific (Waltham, MA, USA). Cells were seeded in 96-well plates at a density of 1 × 10^4^ cells/well. When cells reached 60−70% confluence, cells were given either the 20 µM suffruticosol C treatment or not. Next, 10 µM EdU was added, and cells were added in each well for 2 h. After incubation, the media were removed and 80 µL of 3.7% formaldehyde was added (3.7% in PBS solution) in PBS for 15 min and maintained at room temperature for 15 min. The fixative was removed and the cells were washed twice with 80 µL of 3% BSA (3% in PBS solution). The wash solution was removed, and Triton X-100 (0.3% in PBS solution) was added for 20 min. Sequentially, the endogenous peroxidase’s activity was blocked before EdU detection.

### 2.5. Cell Cycle Analysis

The cell cycle analysis protocol was conducted as outlined in a previous study [35]. Cells were plated in a 6-well plate at a density of 1 × 10^6^ cells/well. When cells reached 60−70% confluence, cold PBS was used to wash the collected cells that were treated with or without 20 μM suffruticosol C for 24 h. The cells were then fixed in 75% ethyl alcohol and kept at 4 °C  for 24 h. Sequentially, a Cell Cycle and Apoptosis Analysis Kit (C6031, BI-OSCIENCE, Shanghai, China) was used to stain the fixed cells. Cell cycle analyses were performed on a flow cytometry (BD FACSAria^TM^ III, San Jose, CA, USA).

### 2.6. Ferroptosis Analysis

Ferroptosis analyses were performed by monitoring the viability of cells (50−60% confluence) that were treated with various concentrations of suffruticosol C. To reduce the impact of additional forms of programmed cell death, cells were treated with the ferroptosis initiator, RSL3, or erastin coupled with suffruticosol C in a brief period of time.

### 2.7. Autophagy Analysis

An autophagy assay was conducted according to the previous description [36]. Cells were plated in a 24-well plate at a density of 3 × 10^5^ cells/well and reached 70−80% confluence before suffruticosol C treatment. Briefly, the GFP-LC3 plasmid was used to transfect the H1299 cells, which were then treated with 20 µM of suffruticosol C for 24 h. Paraformaldehyde (4%) was used to fix the cells, and it was maintained for 15 min. Cell permeabilization was performed using Triton X-100 (0.5% in PBS solution) and kept for another 15 min. Sequentially, it was blocked at room temperature using 1% BSA for 1 h and then stained with DAPI. Finally, the GFP-LC3-containing puncta were measured using a laser scanning confocal microscope (Leica, Wetzlar, Germany).

### 2.8. Cell Apoptosis

The cells were plated in a 6-well plate at a density of 1 × 10^6^ cells/well. When cells reached 70−80% confluence, 20 µM suffruticosol C was given to HT29 and HCT116 cells for a 24 h incubation. The cells were then stained using the YF^®^488-Annexin V/PI apoptosis Detection Kit, after which flow cytometry (BD FACSAria^TM^ III, San Jose, CA, USA) was employed to measure apoptotic cells.

### 2.9. Cell Viability Assay

The cells were seeded in 96-well plates at a density of 1 × 10^4^ cells/well. The cells are on a complete medium containing different concentrations of resveratrol or suffruticosol C for 72 h. To replace the original medium, 100 µL of fresh medium containing 10% CCK-8 reagent was given and kept for a 3 h incubation at a temperature of 37 °C. Finally, the absorbance of each well was read at 450 nm.

### 2.10. Western-Blotting (WB)

The Western-blotting protocol was the same as outlined in our previous study [37]. Briefly, cells were plated in a 24-well plate at a density of 3 × 10^5^ cells/well and reached 70−80% confluence before suffruticosol C treatment. After different concentrations of suffruticosol C treatments for 24 h, the collected cells were washed with PBS and suspended in a Radio Immune Precipitation Assay (RIPA) buffer on ice for 30 min. Proteins were separated using SDS-PAGE and transferred onto nitrocellulose (NC) membranes (0.45 µM, GE). The NC membrane was exposed to the primary antibody for an overnight incubation, followed by three PBS washes. Sequentially, the membranes were then exposed to a horseradish peroxidase (HRP)-conjugated secondary antibody for an incubation of 2 h at room temperature and washed with PBST three times. Finally, the proteins were detected and quantified using the Imaging System (Bio-Rad, Hercules, CA, USA) and ImageJ software (National Institutes of Health, Silver Springs, MD, USA), respectively.

### 2.11. Statistical Analysis

GraphPad Prism 9.0 software (GraphPad Software, La Jolla, CA, USA) was conducted for data analysis. Statistical analyses, including the unpaired two-tailed Student’s *t*-test and a one-way or two-way analysis of variance (ANOVA), were conducted with a *p* value at the 0.05 level. *p* values less than 0.05, 0.01, and 0.001 are marked with *, **, and *** in the graphed data, respectively.

## 3. Results

### 3.1. Effect of Suffruticosol C on the Growth of Various Cancer Cells

Since resveratrol was established as an anti-cancer drug, it was used as a comparator group. The results showed that resveratrol suppressed the cell viability in the colon cancer cell lines HCT116 and HT29 (Figure 1A,B), and the cell viability was reduced by more than 50% when resveratrol’s concentration reached 100 µM (Figure 1A). To investigate whether suffruticosol C had any effect on cellular functions, we evaluated the growth rate of suffruticosol C-treated cells belonging to the colon cancer cell line HCT116 (Figure 1C). The results of our cell proliferation assay demonstrated that suffruticosol C significantly inhibited HCT116 cell growth, and the cell viability was reduced by more than 50% when the suffruticosol C concentration reached 20 µM (Figure 1C). Additionally, we investigated the cell viability of other cancer cells treated by suffruticosol C, such as colon cancer cells (HT29 and Caco2), lung cancer cells (H1299), prostate cancer cells (PC3), and primary hepatocellular carcinoma cells (HepG2). The results confirmed that suffruticosol C remarkably affected the cell viability of these cancer cell lines as well (Figure 1D–H). Thus, our findings suggest that suffruticosol C is more potent than resveratrol in inhibiting the cell viability of diverse types of cancer cells.

### 3.2. Apoptosis-Independent Inhibitory Property of Suffruticosol C on the Growth of Various Cancer Cells

It is widely known that autophagy, apoptosis, and ferroptosis are major mechanisms involved in cell viability. Therefore, the function of suffruticosol C on apoptosis in HT29 and HCT116 cells by FCAS was evaluated. After treatments with 20 µM suffruticosol C, we did not observe an obvious function on the apoptosis of H1299 cells (Figure 2A,B). To confirm that suffruticosol C did not affect the apoptosis of cancer cells, we further investigated the levels of apoptotic proteins of the caspase family and the Bcl-2 family. Caspase 3 is considered to be the most important caspase in the induction of apoptosis and is, therefore, used as an apoptotic marker. Accordingly, the level of cleaved caspase 3 was assessed in cancer cells after treatment with various concentrations of suffruticosol C. As shown in Figure 2C,D, there was no detectable difference in apoptosis rate between the control and suffruticosol C-treated Caco2 and H1299 cells. Moreover, suffruticosol C did not have a significant effect on the anti-apoptotic protein Bcl-2 and pro-apoptotic protein Bax in HT29 and HCT116 cells (Figure 2E,F).

### 3.3. Ferroptosis-Independent Inhibitory Property of Suffruticosol C on the Growth of Various Cancer Cells

Next, we investigated whether suffruticosol C affects sensitivity to ferroptosis initiated by RSL3 or erastin in HT29 and H1299 cells. First, we measured the viability of HT29 cells that had been treated for six hours with various concentrations of RSL3. The viability of cells treated with RSL3 or erastin alone was not significantly different from that of cells treated with the suffruticosol C and RSL3 or the suffruticosol C and erastin combination (Figure 3A,B). Similar to these findings in HT29 cells, suffruticosol C also did not affect the sensitivity of H1299 cells to RSL3/erastin-initiated ferroptosis (Figure 3C,D). These results suggest that cancer cell death induced by suffruticosol C was not dependent on ferroptosis.

### 3.4. Autophagy-Dependent Induction of Cell Death by Suffruticosol C

The treatment of cancer cells with suffruticosol C promoted autophagy, because an increasing number of punctate GFP-LC3 foci were observed after bafilomycin A1 treatment in HT29 and HCT116 cells (Figure 4A–D). Moreover, we investigated the autophagy markers LC3II and p62. A sharp increasing level of LC3II and a sharp decreasing level of p62 were observed in response to treatments with different concentrations of suffruticosol C (Figure 4E–G). This indicates that suffruticosol C mainly affects cell viability through its effects on autophagy. Therefore, we further investigated the function of suffruticosol C in regulating the genes associated with lysosome biogenesis and the restoration of autophagosome initiation. We found that the expression of *TPP1*, *SGSH*, *SCPEP1*, *CSTB*, *ATP6V1H*, *CTSA*, *CTSF*, and *GLA* was significantly enhanced after 24 h treatments with suffruticosol C (Figure 4H). This indicates that autophagy is essential for suffruticosol C-initiated cell death.

### 3.5. Induction of Cell Death by Suffruticosol C via Cell Cycle Arrest

To determine the possible mechanism of suffruticosol C-induced cell growth inhibition in cancer cells, we analyzed the function of suffruticosol C on cell cycle progression. HT29 and HCT116 cells were examined after suffruticosol C exposure with EdU, which is a marker of cell division. Compared with the controls, we observed that the dividing cells significantly decreased after exposure to suffruticosol C for 24 h (Figure 5A,B). To confirm that suffruticosol C interrupted cell division, HepG2 and PC3 cells were treated with suffruticosol C for 24 h, and flow cytometry was employed for analyzing the distribution of cells across cell cycle phases. The results exhibited that the percentage of cells in the G2/M phase increased after suffruticosol C treatments of four cancer cell lines: from 2.13% to 30.4% in HepG2 cells (Figure 5C) and from 3.02% to 30.8% in PC3 cells (Figure 5D). This change was accompanied by an increment in the proportion of suffruticosol C-treated cells in the S phase: from 21.8% to 10.3% in HepG2 cells (Figure 5C) and from 33.5% to 12.8% in PC3 cells (Figure 5D). Overall, these findings suggest that suffruticosol C inhibits cell cycle progression by arresting cells in the G2/M phase.

### 3.6. Inhibition of mTORC1 Activation by Suffruticosol C

To investigate the potential regulatory function of suffruticosol C on mTORC1 activation, we treated H1299 cells with suffruticosol C for 24 h. pT389-S6K or S6, the well-characterized phosphorylation sites of mTORC1, was monitored. We observed that sufruticosol C inhibited the activation of mTORC1 in a dose-dependent manner on activation of mTORC1 inhibited as detected by the phosphorylation of S6K1 and S6 (Figure 6A), as well as in Caco2, H1299, PC3, and HepG2 cells (Figure 6B–F). Based on these findings, we speculated that mTORC1 is involved in suffruticosol C-induced cell death.

### 3.7. Suffruticosol C-Induced Autophagy and Cell Proliferation via Inhibition of the mTORC1 Pathway

To examine whether the function of suffruticosol C on autophagy occurs in an mTORC1-dependent manner, rapamycin, an inhibitor of mTORC1, was used to treat the cells. The results showed that the suffruticosol C treatment did not promote rapamycin-mediated autophagy, as indicated by the presence of GFP-LC3II puncta (Figure 7A,B). Moreover, we examined whether mTORC1 associated with the regulation of cell proliferation mediated by suffruticosol C. Our EdU assay exhibited that rapamycin treatments entirely reversed the effect of suffruticosol C on cell proliferation (Figure 7C). Based on all these findings, we concluded that suffruticosol C induced cell autophagy and cell proliferation by the negative regulation of mTORC1 activities.

## 4. Discussion

In this study, we determined that autophagy and the cell cycle were targeted by suffruticosol C, which is proved by the fact that autophagy and cell cycle arrests are effectively increased by suffruticosol C in various human cancer cells. With regard to the underlying mechanism, we found that the effects of suffruticosol C are dependent on mTORC1. Thus, our findings demonstrate that suffruticosol C has potential anti-cancer effects.

Resveratrol is widely regarded as a promising natural anticancer agent because of its remarkable inhibitory activity against cancer cells [38]. Considerable attention focused on the development or identification of newly derived resveratrol to improve its bio-efficacy and bioavailability [39]. Suffruticosol C is one of the most dominant stilbenes in peony seeds [10,11]. In the present study, we investigated the bioactivity of peony-isolated suffruticosol C, discovered the antitumor activity, including colon cancer cells (HT29, HCT116, and Caco2), lung cancer cells (H1299), prostate cancer cells (PC3), primary hepatocellular carcinoma cells (HepG2), and dissected the underlying mechanism of suffruticosol C against cancer. According to our results, we found that suffruticosol C has a better inhibitory effect on cell viability than resveratrol at the same concentration. A large number of studies suggested that resveratrol plays a crucial role in autophagy, cell cycle, ferroptosis, and apoptosis [24,40,41,42,43,44,45,46,47,48]. However, in our study, we found that suffruticosol C, unlike resveratrol, only has a specific regulatory effect on autophagy and the cell cycle, but it has no effect on apoptosis and ferroptosis. These findings provide the protentional utilization of suffruticosol C in autophagy and cell-cycle-related cancer.

Existing reports have demonstrated that diseases, such as cancer, inflammation, and neurodegenerative diseases, are closely associated with autophagy dysregulation [49]. At the early stage of cancer formation, autophagy can eliminate cancer cells. Activated autophagy, however, can shield tumor cells from injury by chemotherapy or radiation therapy at the late stages of tumor formation, which results in therapy failure [50]. However, an increasing number of studies have proved that activating excessive autophagy can lead to the mass cell death of malignant tumors. Taking 6-bromo-5-trans-4-methoxy benzaldehyde as an example, it has a broad-spectrum effect on carcinoma cell death by inducing autophagy [51]. It was demonstrated that T-oligos could remarkably promote LC3-ll levels in malignant glioma cells and active autophagy by inhibiting mTOR and STAT3 signaling pathways, and then lead to the autophagic death of tumor cells [52]. The anticancer drug tamoxifen can upregulate the expression level of Beclin1 to induce autophagy [52]. Rapamycin, known as an autophagy inducer, has entered phase Ⅲ of clinical trials [53]. Therefore, the application of autophagy inducers to stimulate excessive autophagy in tumor cells may become a novel idea for future cancer prevention and treatment, and the autophagy induction of suffruticosol C in a variety of malignant cell lines may exert its anticancer activity.

The mTORC1 regulatory network has drawn attention for its important role in various disorders [54]. Studies have shown that mTORC1 activation can be induced by various amino acids, growth factors, and also glucose [54]. In addition, some studies reported that resveratrol may also regulate mTORC1 activations [27,28,29,30]. Accordingly, in the current study, we found that suffruticosol C can effectively regulate the activation of mTORC1. Our data showed that suffruticosol C treatments could inhibit the phosphorylation levels of S6K1. Many environmental cues, including amino acids, growth factors, cholesterol, lipids, and glucose can influence the activation of mTORC1 [55]. However, in our study, we only examined the effect of suffruticosol C on the baseline activation of mTORC1. We did not investigate which signal-mediated mTORC1 pathway (glucose, growth factors, cholesterol, lipid, or amino acid) is specifically inhibited by suffruticosol C, so this requires further investigation.

Our findings suggest that suffruticosol C-mediated induction autophagy and cell cycle arrest occurred via the inhibition of the mTORC1 pathway, as the function of suffruticosol C on autophagy and cell cycle was completely abrogated in rapamycin-treated cells. Although our results demonstrated that suffruticosol C can regulate cell cycle progression via its effect on mTORC1 activation, we did not investigate the details of this molecular mechanism. Published works have suggested that the inhibition of mTORC1 mainly leads to cell cycle arrest in the G1/S phase [25,26], while in our work, we observed that suffruticosol C causes cell cycle arrest mainly in the G2/M and S phases. Numerous studies have shown that 14-3-3σ, cdc25C, and microRNA mir34a are essential in arresting cells in the G2/M phase [56,57,58,59], so future studies could explore whether suffruticosol C results in cell growth arrest by controlling the activity or expression of these factors. Accordingly, in future studies, we plan to confirm the present findings via in vivo experiments using mouse models, including xenograft tumors. As mTORC1 plays a critical role in many serious diseases, particularly cancer, our findings provide an insightful understanding of the pathogenesis and progress of various diseases.

## 5. Conclusions

In summary, we used a variety of biochemical and cell-based methods to demonstrate the effects of suffruticosol C on cancer cell viability, and our results demonstrate that it affects autophagy and the cell cycle’s progression in cancer cells via its regulatory effects on the mTORC1 pathway (Graphical abstract). The suffruticosol C-mediated mTORC1 inactivation mechanism elucidated in this study may be useful for identifying therapeutic targets for cancer treatments in the future.

## Figures and Tables

**Figure 1 nutrients-14-05000-f001:**
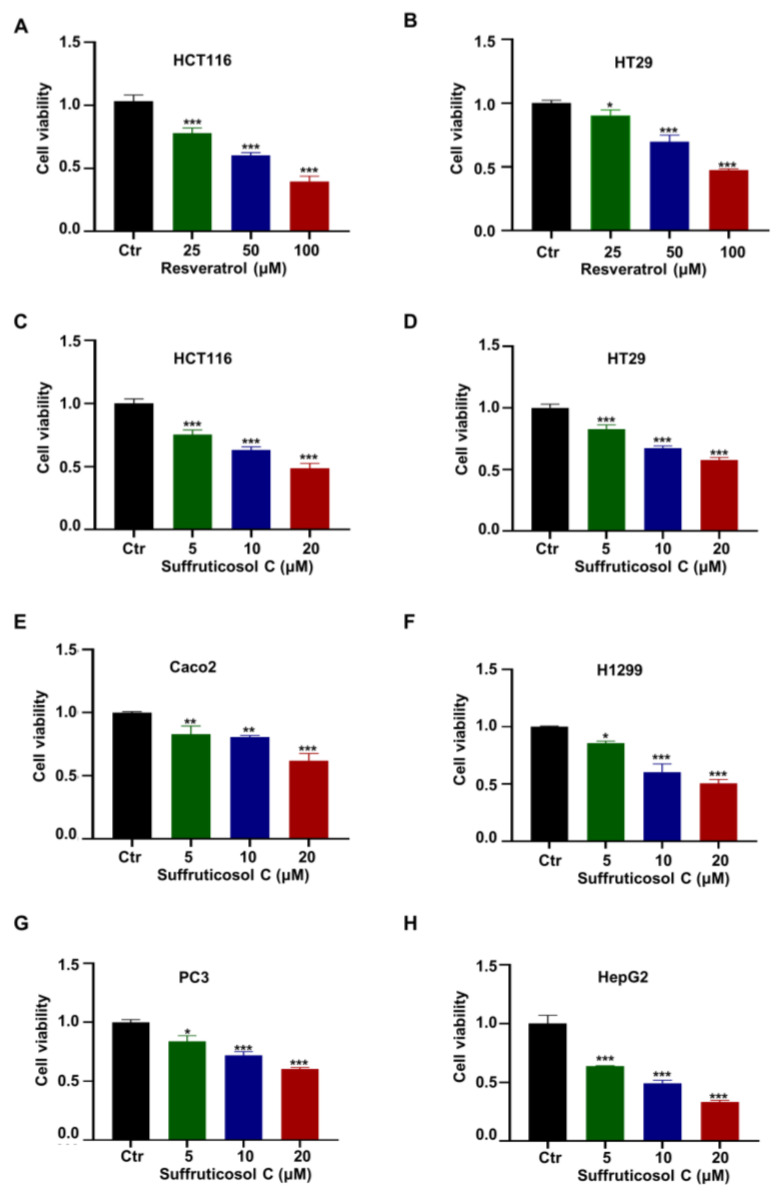
Effect of suffruticosol C and resveratrol on the growth of various cancer cells. HCT116 (**A**) and HT29 (**B**) cells were treated with different concentrations of resveratrol for 72 h, and the viability of cells was detected by CCK-8. HCT116 (**C**), HT29 (**D**), Caco2 (**E**), H1299 (**F**), PC3 (**G**), and HepG2 (**H**) cells were treated with different concentrations of suffruticosol C for 72 h, and the viability of cells was detected by CCK-8. All samples were normalized to cell number and conducted with three independent replicates. Data were analyzed by one-way ANOVA (**A**–**H**) were expressed as Mean ± SD (**A**–**H**). Different colors represent the different concentration of resveratrol or suffruticosol C. *, *p*  <  0.05; **, *p*  <  0.01; ***, *p*  <  0.001. Ctr: normal control group.

**Figure 2 nutrients-14-05000-f002:**
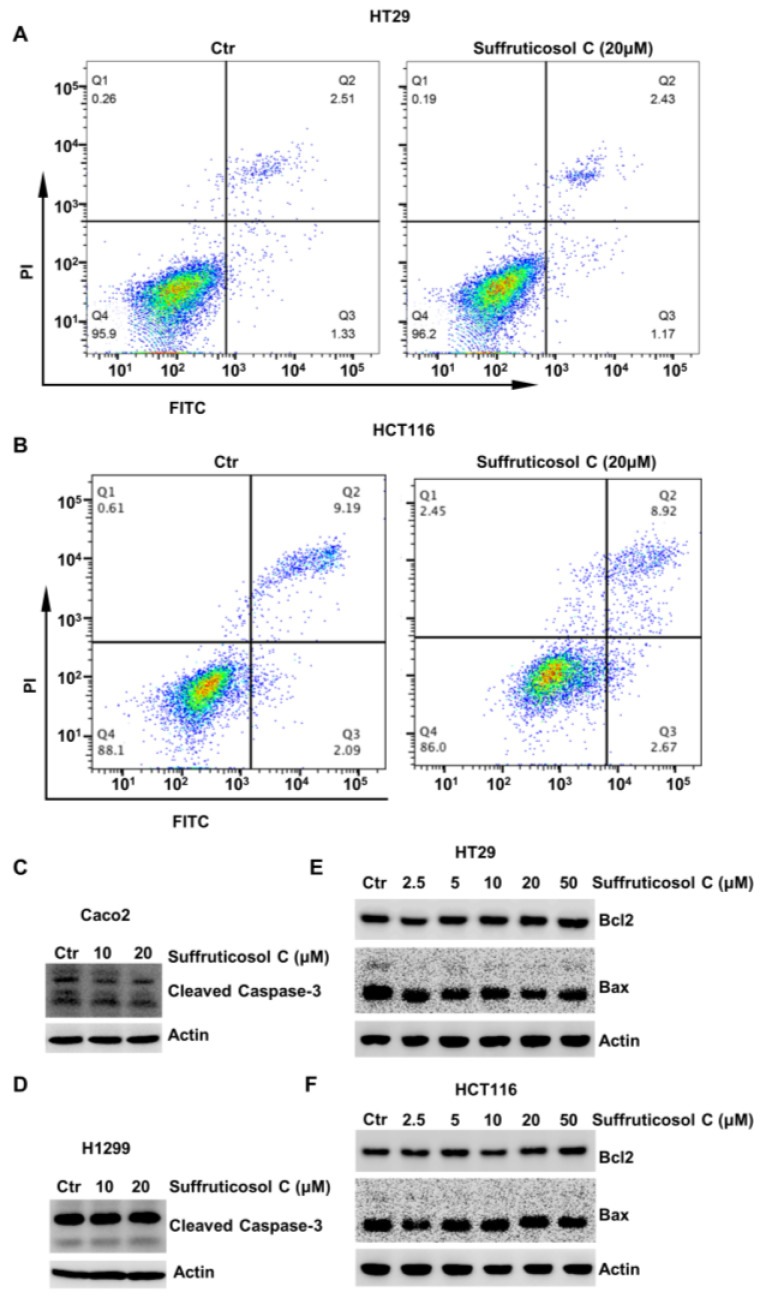
The apoptosis-independent inhibitory effect of suffruticosol C on the growth of various cancer cells. HT29 (**A**) and HCT116 (**B**) cells were treated with suffruticosol C for 24 h, and apoptosis was examined using FACS. Caco2 (**C**) and H1299 (**D**) cells were treated with suffruticosol C for 24 h, and the level of cleaved caspase-3 was detected by WB. HT29 (**E**) and HCT116 (**F**) cells were treated with suffruticosol C for 24 h, and the level of Bcl-2 and Bax were detected by WB.

**Figure 3 nutrients-14-05000-f003:**
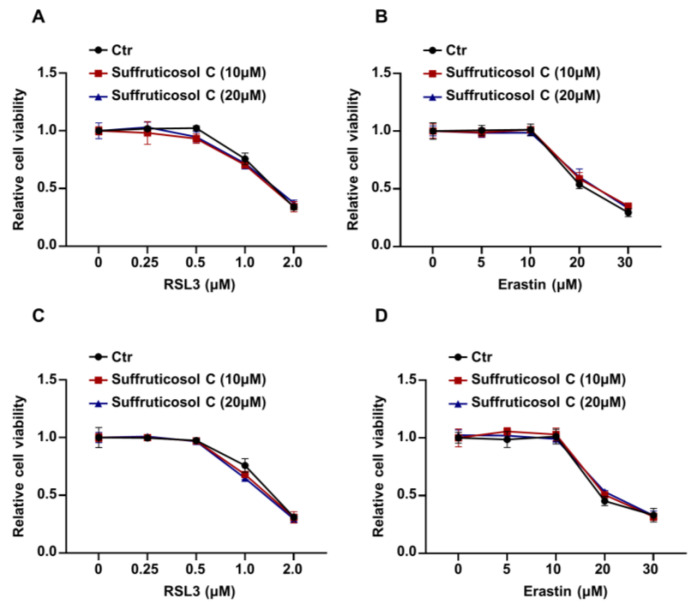
Ferroptosis-independent inhibitory effect of suffruticosol C on the growth of various cancer cells. (**A**,**B**). HT29 cells were treated with different concentrations of RSL3 (**A**) or erastin (**B**) for 24 h in combination with indicated concentrations of suffruticosol C, and the viability of indicated cells was examined using CCK-8. (**C**,**D**). H1299 cells were treated with different concentrations of RSL3 (**C**) or erastin (**D**) for 24 h in combination with indicated concentrations of suffruticosol C, and the viability of indicated cells was examined using CCK-8. Data were analyzed by two-way ANOVA (**A**–**D**) and the results were expressed as Mean ± SD (**A**–**D**).

**Figure 4 nutrients-14-05000-f004:**
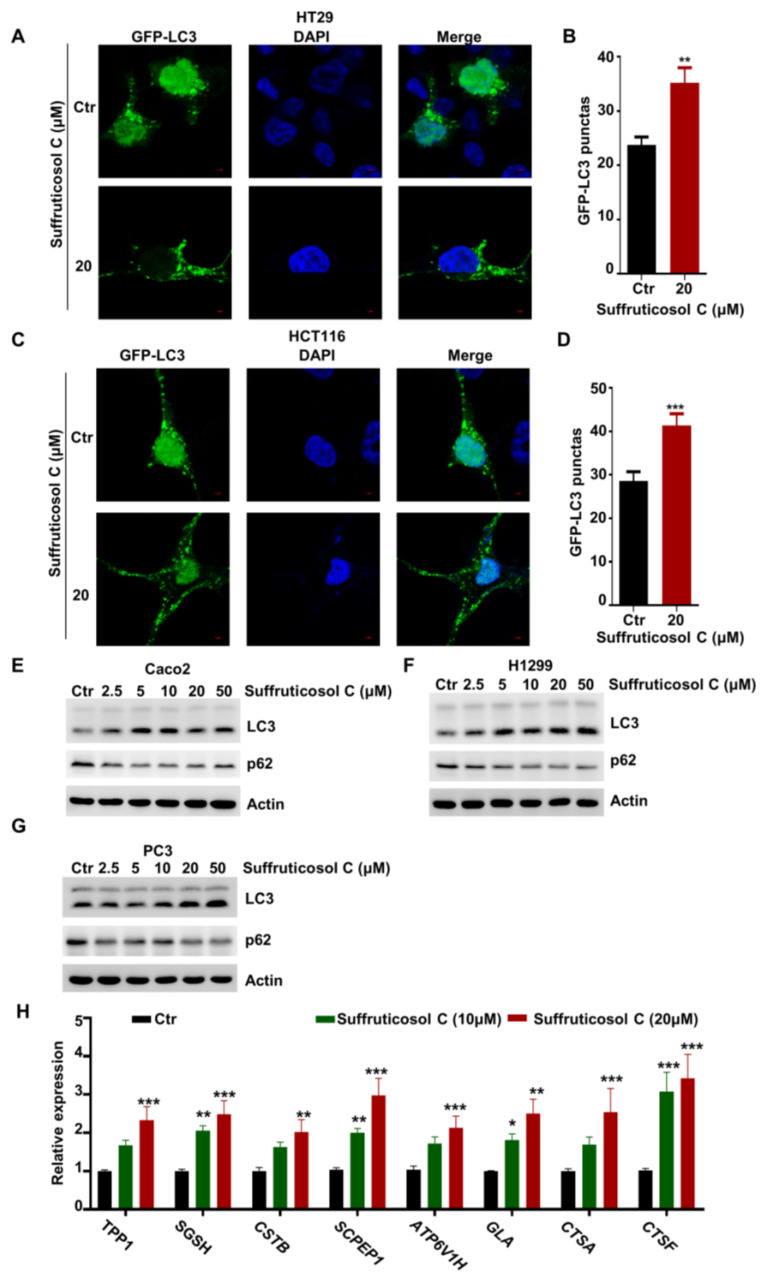
Autophagy-dependent induction of cell death by suffruticosol C. HT29 (**A**,**B**) and H1299 (**C**,**D**) cells were treated with suffruticosol C for 24 h, after which cell autophagy was analyzed by examining GFP-LC3 puncta (**A**,**C**). Quantitative data for the GFP-LC3 puncta are presented (**B**,**D**). Caco2 (**E**), H1299 (**F**), and PC3 (**G**) cells were treated with different concentrations of suffruticosol C for 24 h, after which autophagy was analyzed by examining LC3II and P62 levels. (**H**). HT29 cells were treated with suffruticosol C for 24 h, and the expression of *TPP1*, *SGSH*, *CSTB*, *SCPEP1*, *ATR6V1H*, *GLA*, *GTSA*, and *CTSF* was detected by qRT-PCR. Data were analyzed by unpaired two-tailed Student’s *t*-test (**B**,**D**) or two-way ANOVA (H), and the results were expressed as Mean ± SEM (**B**,**D**) or Mean ± SD (**H**). *, *p* < 0.05; **, *p* < 0.01; ***, *p* < 0.001.

**Figure 5 nutrients-14-05000-f005:**
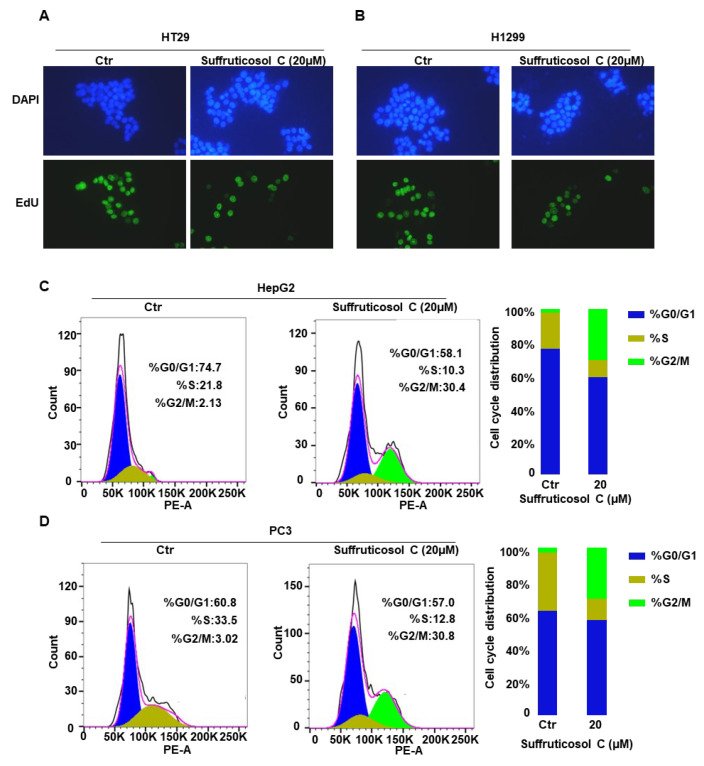
Induction of cell death by suffruticosol C via cell cycle arrest. HT29 (**A**) and H1299 (**B**) cells were treated with suffruticosol C for 24 h, and the cell proliferation analysis was undertaken using an EdU (5-ethynyl-2-deoxyuridine) assay. HepG2 (**C**) and PC3 (**D**) cells were treated with suffruticosol C for 24 h, the cell cycle analysis using flow cytometry after staining with propidium iodide (PI).

**Figure 6 nutrients-14-05000-f006:**
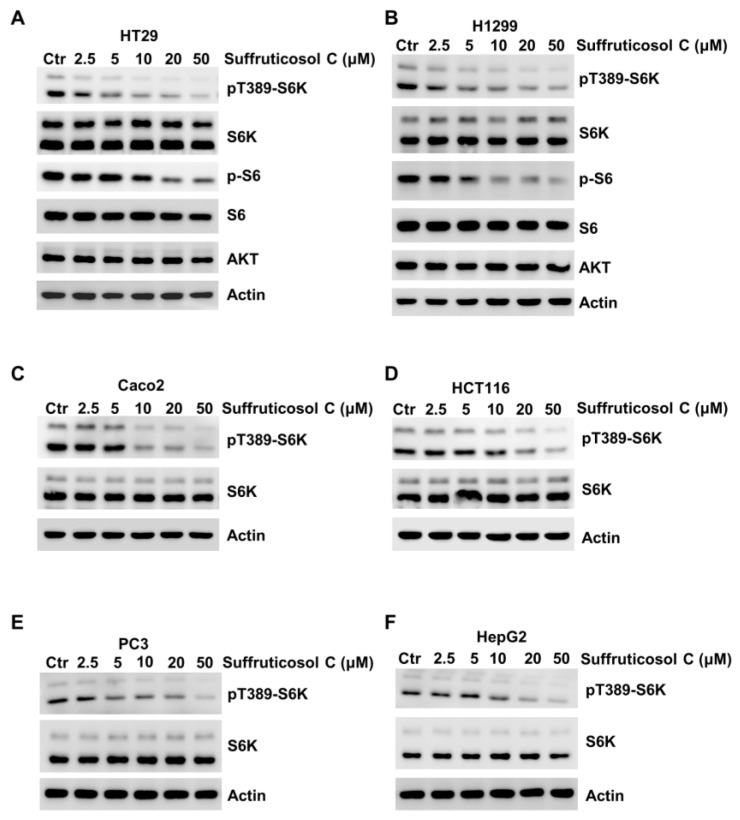
Suffruticosol C inhibits mTORC1 activation. HT29 (**A**), H1299 (**B**), Caco2 (**C**), HCT116 (**D**), PC3 (**E**), and HepG2 (**F**) cells were treated with different concentrations of suffruticosol C for 24 h, and the level of pT389-S6K, p-S6, and the indicated protein were detected by WB.

**Figure 7 nutrients-14-05000-f007:**
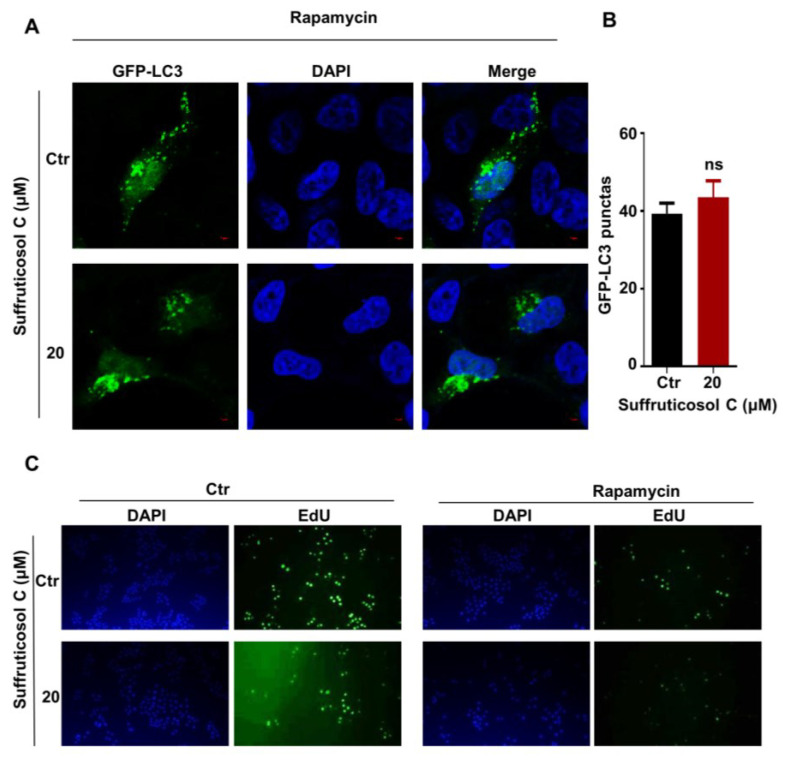
Suffruticosol C-induced autophagy and cell proliferation via inhibition of the mTORC1 pathway. HT29 cells were treated with rapamycin or in combination with suffruticosol C for 24 h, after which autophagy was analyzed by examining GFP-LC3 puncta (**A**). Quantitative data for the GFP-LC3 puncta are presented (**B**). (**C**). HT29 cells were treated with rapamycin or in combination with suffruticosol C for 24 h, after which a cell proliferation analysis was undertaken using an EdU assay. Data were analyzed by unpaired two-tailed Student’s *t*-test (**B**), and the results were expressed as Mean ± SEM (**B**). ns, not significant.

**Table 1 nutrients-14-05000-t001:** Primer sequences for qRT-PCR.

Gene	Forward Primer Sequence (5′→3′)	Reverse Primer Sequence (5′→3′)
*CTSA*	CAGGCTTTGGTCTTCTCTCCA	TCACGCATTCCAGGTCTTTG
*TPP1*	GATCCCAGCTCTCCTCAATACG	GCCATTTTTGCACCGTGTG
*SGSH*	TGACCGGCCTTTCTTCCTCTA	GCTCTCTCCGTTGCCAAACTT
*CSTB*	AGTGGAGAATGGCACACCCTA	AAGAAGCCATTGTCACCCCA
*GLA*	AGCCAGATTCCTGCATCAGTG	ATAACCTGCATCCTTCCAGCC
*SCPEP1*	GATCTCCCCTGTTGATTCGGT	AGCCCCTTATTTACGGCATTC
*CTSF*	ACAGAGGAGGAGTTCCGCACTA	GCTTGCTTCATCTTGTTGCCA
*ATP6V1H*	GGAAGTGTCAGATGATCCCCA	CCGTTTGCCTCGTGGATAAT
*GAPDH*	CAACGAATTTGGCTACAGCA	AGGGGTCTACATGGCAACTG

## Data Availability

Not applicable.

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
