# Peer review of "Suffruticosol C-Mediated Autophagy and Cell Cycle Arrest via Inhibition of mTORC1 Signaling"

_nutrients, 2022, doi:10.3390/nu14235000_

Round 1
Reviewer 1 Report
Summary: The manuscript entitled as “Suffruticosol C-mediated autophagy and cell cycle arrest via inhibition of mTORC1 signaling” is focused on anticancer property of phytoconstituent suffruticosol C.
1. The manuscript is organized but language should be checked one more time, there are some typo errors as well.
2. Authors did not mention the numbers of cells per well and confluency in EdU assay, Cell cycle analysis? It would be useful for the target audience if this information is added into the manuscript.
3. Some of the methods are not presented with the proper references, it is suggested to cite the relevant references.
4. How much quantity of the Suffruticosol C was used to treat the different cell lines in the methods? and on what basis the dose was selected?
5. Figure 1, it is not clear how the data are presented whether its mean ± SEM or mean ± SD? One may get confused. It is suggested to add the information.
What does the “0” µM represent here? Is it a control?
Author Response
Review 1#
Summary: The manuscript entitled as “Suffruticosol C-mediated autophagy and cell cycle arrest via inhibition of mTORC1 signaling” is focused on anticancer property of phytoconstituent suffruticosol C.
Response: We sincerely thank the reviewer for thoroughly examining our manuscript and providing very helpful comments to guide our revision. We have tried our best to revise the manuscript according to the reviewer's kind and construction comments and suggestions. We sincerely hope that this revised manuscript has addressed all the comments and suggestions.
1. The manuscript is organized but language should be checked one more time, there are some typo errors as well.
Response: Thanks for the reviewer's insightful suggestion. We have checked and revised the manuscript carefully and revised some errors including the typo errors in it as well. Moreover, the manuscript is polished and modified by experts.
2. Authors did not mention the numbers of cells per well and confluency in EdU assay, Cell cycle analysis? It would be useful for the target audience if this information is added into the manuscript.
Response: Thanks to the reviewer for pointing this out, and we feel sorry that we did not mention the numbers of cells per well and confluency in EdU assay and Cell cycle analysis. Indeed, cells were plated in a 96-well plate at a density of 1 × 104 cells/well, and cells reached 60−70% confluency before EdU assay. In cell cycle analysis, cells were plated in a 6-well plate at a density of 1 × 106 cells/well and reached 60−70% confluency. We have added the numbers of cells per well and confluency to the materials and methods in the revised manuscript (Lines 112-113, 123-124).
3. Some of the methods are not presented with the proper references, it is suggested to cite the relevant references.
Response: Thanks for the reviewer’s insightful comments and we are sorry that some of the methods are not presented with the proper references. Briefly, the quantitative RT-PCR analysis refered to Livak’s et al. work (PMID: 11846609), the EdU assay refered to Salic’s et al. work (PMID: 18272492) and the cell cycle assay refered to Wu’s et al work (PMID: 31569380). We have cited the relevant references to the quantitative RT-PCR analysis, EdU assay and cell cycle assay in the materials and methods of the revised manuscript (Lines 106, 111, 122).
4. How much quantity of the Suffruticosol C was used to treat the different cell lines in the methods? and on what basis the dose was selected?
Response: Thanks for the reviewer's constructive suggestions, and we apologize for the confusion for the dose of suffruticosol C. In Figure 1, we used 20 µM of suffruticosol C as the highest concentration for treatment of HCT116, Caco2, and HepG2 cell lines. HT29, H1299 and PC3 cell lines was treated with a maximum 50 µM concentration of suffruticosol C. However, to examine the effect on different cell lines at the same concentration, we used 20 μM suffruticosol C to treat the different cell lines in the EdU assay, cell cycle analysis, autophagy analysis and cell apoptosis analysis. We have added the quantity of the suffruticosol C used to treat the different cell lines in the materials and methods of the revised manuscript (Lines 125, 137-138, 145, 158).
Our choice of the concentration of suffruticosol C is based on some literature researches. For example, Marlies, Ruye and Paula et al (PMID: 29706321; PMID: 33891090; PMID: 19108833) used 50 or 100 µM resveratrol concentration to treat cells. And considering that suffruticosol C is a trimer of resveratrol, so we carried out the experiments according to 1/3 of the dose of resveratrol. In addition, Jin Hu et al (PMID: 25629409) used 20 µM concentration of suffruticosol C to treat N2a cells, so we used concentration of 20 or 50 µM to treat different cancer cell lines, and we added this information to the revised manuscript (Lines 84-86).
5. Figure 1, it is not clear how the data are presented whether its mean ± SEM or mean ± SD? One may get confused. It is suggested to add the information.
Response: Thanks for the reviewer’s reminding, and we feel very sorry for the confusion that it is not clear how the data are presented. We have complemented the information of mean ± SEM or mean ± SD in figure legends of the revised manuscript (Lines 195-196, 235-236, 257-259, 308-310).
What does the “0” µM represent here? Is it a control?
Response: Thanks to the reviewer for pointing this out. We apologized for the confusion of “0” µM and “control” here. In fact, the “0” µM represents a control. In order not to cause misunderstanding, we have replaced “0” µM with “Ctr” in all figures of the revised manuscript.
Reviewer 2 Report
The authors demonstrated that suffruticosol C induces autophagy and cell cycle arrest via inhibiting mTORC1 signaling. I think the results presented are adequate to justify its publication. My major comments would be the lack of a comparator group of resveratrol, wherein the anticancer is more established. The discussion should be more detailed also, contrasting the effects of suffruticosol C with other families of stilbenes. Please also add a schematic diagram to illustrate the mechanism of suffruticosol C as well.
Minor comments:
line 14: delete "too"
Methodology: Clearly indicate the concentration range and cell density for each experiment.
line 120: do not start the sentence with a numerical number.
Statistical test: when was the one-tailed test used?
Figure 4H: please correct the spelling of the relative expression.
Author Response
Review 2#
The authors demonstrated that suffruticosol C induces autophagy and cell cycle arrest via inhibiting mTORC1 signaling. I think the results presented are adequate to justify its publication.
Response: We thank the reviewer for reading our manuscript carefully and giving the positive recognition and comments. We have performed related experiments to address the concerns of the reviewer. We sincerely hope that this revised manuscript has addressed all the reviewer's comments and suggestions.
My major comments would be the lack of a comparator group of resveratrol, wherein the anticancer is more established.
Response: Thanks for the reviewer's constructive suggestion. According to the reviewer's suggestions, we added new experiments: in the cell viability assay, HCT116 and HT29 cells were treated with different concentrations of resveratrol as a comparator group. We found that resveratrol significantly reduced the growth rate of cells and the cell viability was reduced by more than 50% when the resveratrol concentration reached 100 µM (Figure 1A & B). However, the suffruticosol C markedly inhibits the growth rate of HCT116 cells, and the cell viability was reduced by more than 50% when the suffruticosol C concentration reached 20 µM (Figure 1C). Thus, our findings suggest that suffruticosol C is more potent than resveratrol in inhibiting cell viability of various types of tumor cells. We have added these contents to the results in the revised manuscript (Lines 174-177, 187-188).
The discussion should be more detailed also, contrasting the effects of suffruticosol C with other families of stilbenes.
Response: Thanks for the reviewer's insightful comments, and the comparison of the antitumor effects of suffruticosol C with resveratrol was analyzed in the discussion (Lines 318-331).
Resveratrol is widely regarded as a promising natural anticancer agent because of its remarkable inhibitory activity against cancer cells (PMID: 11406544). Considerable attention focused on the development or identification of newly derived resveratrol to improve their bio-efficacy and bioavailability (PMID: 25862967). Suffruticosol C is one of the most dominant stilbenes in peony seeds (PMID: 25532833; PMID: 20522997). In present study, we investigated the bioactivity of peony-isolated suffruticosol C, discovered the antitumor activity, including colon cancer cells (HT29, HCT116, and Caco2), lung cancer cells (H1299), prostate cancer cells (PC3), primary hepatocellular carcinoma cells (HepG2), and dissected the underlying mechanism of suffruticosol C against cancer. Previous studies suggested that resveratrol plays a crucial role in autophagy, cell cycle, ferroptosis, and apoptosis (PMID: 35277058; PMID: 32896720; PMID: 34530090; PMID: 35685642). However, in our study, we found that suffruticosol C, unlike resveratrol, only has a specific regulatory effect on autophagy and the cell cycle but it has no effect on apoptosis and ferroptosis. These findings provide the protentional utilization of suffruticosol C in autophagy and cell-cycle-related cancer.
Please also add a schematic diagram to illustrate the mechanism of suffruticosol C as well.
Response: Thanks for the reviewer's constructive suggestion. In order to give the reader a better understanding of the mechanism of suffruticosol C, we have added a schematic diagram (Graphical Abstract) below the abstract (Line 23).
Minor comments:
line 14: delete "too"
Response: We thank the reviewer for pointing this out and are sorry for this mistake. We have deleted "too" in line 14 of the revised manuscript.
Methodology: Clearly indicate the concentration range and cell density for each experiment.
Response: Thanks for the review's insightful comment, and we feel sorry that we did not clearly indicate the concentration range and cell density for each experiment. In EdU assay, cells were plated in a 96-well plate at a density of 1 × 104 cells/well and when cells reached 60−70% confluence, cells were treated with or without 20 μM suffruticosol C for 24 h before EdU assay. In cell cycle analysis, cells were plated in a 6-well plate at a density of 1 × 106 cells/well and when cells reached 60−70% confluence, cells were treated with or without 20 µM suffruticosol C for 24 h. In Western-blotting assay, cells were plated in a 24-well plate at a density of 3 × 105 cells/well and when cells reached 70−80% confluence, cells were treated with 20 µM suffruticosol C for 24h. We have added the concentration range and cell density for each experiment in the materials and methods of the revised manuscript (Lines 112-113, 123-125, 135-137, 144-145, 150, 157-158).
line 120: do not start the sentence with a numerical number.
Response: Thanks to the reviewer very much for pointing this out and we are very sorry for our presentation inattention. We have corrected the sentence to “Cells were seeded in 96-well plates at a density of 1 × 104 cells/well” in Line 123 of the revised manuscript.
Statistical test: when was the one-tailed test used?
Response: We thank the reviewer for pointing this out and are really sorry for the error and confusion we have caused. In our study, the statistical analysis we used includes unpaired two-tailed Student's t-test, one-way ANOVA and two-way ANOVA. In our revised manuscript, we have corrected the statistical test in Lines 169-170 and illustrated in the legend exactly the statistical analysis used for each figure (Lines 195-196, 235-236, 257-259, 308-310).
Figure 4H: please correct the spelling of the relative expression.
Response: Thanks to the reviewer very much for pointing this out and we are very sorry for our presentation mistakes. We have corrected “Relativize expression” to “Relative expression” in Figure 4H of the revised manuscript.
Round 2
Reviewer 2 Report
Dear authors,
Thanks for addressing my comments. I have some further queries:
1. Is the schematic diagram presented at the beginning of the manuscript the graphical abstract? It is not cited in the text and the description is not provided as footnotes.
2. In the experiments presented in Figure 1, why some tests were performed as time-course and some end-point? Please amend the title as well because the tests involve both suffruticosol C and resveratrol.
3. For the subsequent mechanistic study of cell death, there is no resveratrol group, which is a major limitation.
Author Response
Dear Kristen Gao,
This manuscript is a revised and extended version of manuscript (No.: nutrients-2035289) entitled “Suffruticosol C-mediated autophagy and cell cycle arrest via inhibition of mTORC1 signaling” by Qin et al. Thank you very much for editing and advice on how to improve our manuscript. We sincerely thank the reviewer for thoroughly examining our manuscript and providing very helpful comments to guide our revision. We have tried our best to revise the manuscript according to the reviewer's kind and construction comments and suggestions. We sincerely hope that our revised manuscript has addressed all the comments and suggestions.
Comments and suggestions of Reviewer 2:
1. Is the schematic diagram presented at the beginning of the manuscript the graphical abstract? It is not cited in the text and the description is not provided as footnotes.
Response: Thanks to the reviewer for pointing this out, and we feel sorry that we did not cite and descript the graphical abstract in the text. As the reviewer said, the schematic diagram presented at the beginning of the manuscript is indeed a graphical abstract. We have cited the graphical abstract in Line 384 of the revised text and the description “Proposed a mechanism by which suffruticosol C regulates cancer cell viability via the inhibition of mTORC1 signaling.” of the graphical abstract has been provided as footnotes in Lines 24-25.
2. In the experiments presented in Figure 1, why some tests were performed as time-course and some end-point? Please amend the title as well because the tests involve both suffruticosol C and resveratrol.
Response: Thanks for the reviewer's constructive suggestions, and we apologize for the confusion for the time of suffruticosol C treatment. In order to further eliminate reviewer's concerns, the treatment time and concentration of resveratrol and suffruticosol C in Figure 1 have been unified, and 72 h of the treatment time was finally selected. Moreover, according to the reviewer's suggestions, we have amended the title as well to “Effect of suffruticosol C and resveratrol on the growth of various cancer cells” in Line 192.
3. For the subsequent mechanistic study of cell death, there is no resveratrol group, which is a major limitation.
Response: Thanks for the reviewer's constructive suggestion. In fact, resveratrol was used as a positive control in present work. The effects of resveratrol on cell autophagy [1-9], cell cycle [10-20], and cell apoptosis [12, 17, 20-23] have been widely reported in a large number of literatures. We have conducted relevant cell viability experiments of suffruticosol C and resveratrol (Figure 1A & B). According to our results, we found that suffruticosol C, a trimer of resveratrol, has a better inhibitory effect on cell viability than resveratrol at the same concentration (Figure 1), so the subsequent experiments mainly focus on suffruticosol C. We understand that adding resveratrol group for the subsequent mechanistic study of cell death may better reveal the antitumor activity of suffruticosol C. However, in the present study, we mainly focused on suffruticosol C, and we think that it is not necessary to include resveratrol group, which may weaken the study topic. We sincerely apologize for having the difference with the reviewer on this point and we hope such explanation could be acceptable for reviewer. In consideration of the reviewer's concerns, we have added these contents to the discussion of our revised manuscript (Lines 328-331).
Best regards.
Sincerely yours,
Lu Deng, Ph.D. Professor, College of Animal Science and Technology,
Northwest A&F University,
Yangling Shaanxi 712100, China
Email: [email protected]
References
[1] Marino, G., E. Morselli, M.V. Bennetzen, T. Eisenberg, E. Megalou, S. Schroeder, S. Cabrera, P. Benit, P. Rustin, A. Criollo, O. Kepp, L. Galluzzi, S. Shen, S.A. Malik, M.C. Maiuri, Y. Horio, C. Lopez-Otin, J.S. Andersen, N. Tavernarakis, F. Madeo, G. Kroemer, Autophagy, 2011, 7, 647-9.
[2] Selvaraj, S., Y. Sun, P. Sukumaran, B.B. Singh, Mol Carcinog, 2016, 55, 818-31.
[3] Garcia-Zepeda, S.P., E. Garcia-Villa, J. Diaz-Chavez, R. Hernandez-Pando, P. Gariglio, Eur J Cancer Prev, 2013, 22, 577-84.
[4] Ma, R., D. Yu, Y. Peng, H. Yi, Y. Wang, T. Cheng, B. Shi, G. Yang, W. Lai, X. Wu, Y. Lu, J. Shi, Acta Biochim Biophys Sin (Shanghai), 2021, 53, 775-783.
[5] Fan, Y., J.F. Chiu, J. Liu, Y. Deng, C. Xu, J. Zhang, G. Li, BMC Cancer, 2018, 18, 581.
[6] Andreadi, C., R.G. Britton, K.R. Patel, K. Brown, Autophagy, 2014, 10, 524-5.
[7] Li, J., Y. Fan, Y. Zhang, Y. Liu, Y. Yu, M. Ma, Nutrients, 2022, 14.
[8] Gurusamy, N., I. Lekli, S. Mukherjee, D. Ray, M.K. Ahsan, M. Gherghiceanu, L.M. Popescu, D.K. Das, Cardiovasc Res, 2010, 86, 103-12.
[9] Mauthe, M., A. Jacob, S. Freiberger, K. Hentschel, Y.D. Stierhof, P. Codogno, T. Proikas-Cezanne, Autophagy, 2011, 7, 1448-61.
[10] Pozo-Guisado, E., A. Alvarez-Barrientos, S. Mulero-Navarro, B. Santiago-Josefat, P.M. Fernandez-Salguero, Biochem Pharmacol, 2002, 64, 1375-86.
[11] Delmas, D., P. Passilly-Degrace, B. Jannin, M. Cherkaoui Malki, N. Latruffe, Int J Mol Med, 2002, 10, 193-9.
[12] Bhardwaj, A., G. Sethi, S. Vadhan-Raj, C. Bueso-Ramos, Y. Takada, U. Gaur, A.S. Nair, S. Shishodia, B.B. Aggarwal, Blood, 2007, 109, 2293-302.
[13] Liang, Y.C., S.H. Tsai, L. Chen, S.Y. Lin-Shiau, J.K. Lin, Biochem Pharmacol, 2003, 65, 1053-60.
[14] Sgambato, A., R. Ardito, B. Faraglia, A. Boninsegna, F.I. Wolf, A. Cittadini, Mutat Res, 2001, 496, 171-80.
[15] Benitez, D.A., E. Pozo-Guisado, A. Alvarez-Barrientos, P.M. Fernandez-Salguero, E.A. Castellon, J Androl, 2007, 28, 282-93.
[16] Zhang, Y., Y. Li, C. Sun, X. Chen, L. Han, T. Wang, J. Liu, X. Chen, D. Zhao, Cancers (Basel), 2021, 13.
[17] Wu, H., L. Chen, F. Zhu, X. Han, L. Sun, K. Chen, Toxins (Basel), 2019, 11.
[18] Jang, J., J. Song, J. Lee, S.K. Moon, B. Moon, Int J Mol Sci, 2021, 22.
[19] Yang, H.Z., J. Zhang, J. Zeng, S. Liu, F. Zhou, F. Zhang, F. Giampieri, D. Cianciosi, T.Y. Forbes-Hernandez, J. Ansary, E. Gil, R. Chen, M. Battino, Int J Food Sci Nutr, 2020, 71, 84-93.
[20] Estrov, Z., S. Shishodia, S. Faderl, D. Harris, Q. Van, H.M. Kantarjian, M. Talpaz, B.B. Aggarwal, Blood, 2003, 102, 987-95.
[21] Sexton, E., C. Van Themsche, K. LeBlanc, S. Parent, P. Lemoine, E. Asselin, Mol Cancer, 2006, 5, 45.
[22] Zhao, S., L. Tang, W. Chen, J. Su, F. Li, X. Chen, L. Wu, Naunyn Schmiedebergs Arch Pharmacol, 2021, 394, 797-807.
[23] Bostan, M., M. Mihaila, G.G. Petrica-Matei, N. Radu, R. Hainarosie, C.D. Stefanescu, V. Roman, C.C. Diaconu, Int J Mol Sci, 2021, 22.